# Clustering Signed Networks with the Geometric Mean of Laplacians

**Pedro Mercado[1], Francesco Tudisco[2] and Matthias Hein[1]**
[1]Saarland University, Saarbrücken, Germany
[2]University of Padua, Padua, Italy

## Abstract

Signed networks allow to model positive and negative relationships. We analyze existing extensions of spectral clustering to signed networks. It turns out that existing approaches do not recover the ground truth clustering in several situations where either the positive or the negative network structures contain no noise. Our analysis shows that these problems arise as existing approaches take some form of arithmetic mean of the Laplacians of the positive and negative part. As a solution we propose to use the geometric mean of the Laplacians of positive and negative part and show that it outperforms the existing approaches. While the geometric mean of matrices is computationally expensive, we show that eigenvectors of the geometric mean can be computed efficiently, leading to a numerical scheme for sparse matrices which is of independent interest.

## 1 Introduction

A signed graph is a graph with positive and negative edge weights. Typically positive edges model attractive relationships between objects such as similarity or friendship and negative edges model repelling relationships such as dissimilarity or enmity. The concept of balanced signed networks can be traced back to [10, 3]. Later, in [5], a signed graph is defined as $k$-balanced if there exists a partition into $k$ groups where only positive edges are within the groups and negative edges are between the groups. Several approaches to find communities in signed graphs have been proposed (see [23] for an overview). In this paper we focus on extensions of spectral clustering to signed graphs. Spectral clustering is a well established method for unsigned graphs which, based on the first eigenvectors of the graph Laplacian, embeds nodes of the graphs in $\mathbb{R}^k$ and then uses $k$-means to find the partition. In [16] the idea is transferred to signed graphs. They define the signed ratio and normalized cut functions and show that the spectrum of suitable signed graph Laplacians yield a relaxation of those objectives. In [4] other objective functions for signed graphs are introduced. They show that a relaxation of their objectives is equivalent to weighted kernel $k$-means by choosing an appropriate kernel. While they have a scalable method for clustering, they report that they can not find any cluster structure in real world signed networks.

We show that the existing extensions of the graph Laplacian to signed graphs used for spectral clustering have severe deficiencies. Our analysis of the stochastic block model for signed graphs shows that, even for the perfectly balanced case, recovery of the ground-truth clusters is not guaranteed. The reason is that the eigenvectors encoding the cluster structure do not necessarily correspond to the smallest eigenvalues, thus leading to a noisy embedding of the data points and in turn failure of $k$-means to recover the cluster structure. The implicit mathematical reason is that all existing extensions of the graph Laplacian are based on some form of arithmetic mean of operators of the positive and negative graphs. In this paper we suggest as a solution to use the geometric mean of the Laplacians of positive and negative part. In particular, we show that in the stochastic block model the geometric mean Laplacian allows in expectation to recover the ground-truth clusters in

any reasonable clustering setting. A main challenge for our approach is that the geometric mean Laplacian is computationally expensive and does not scale to large sparse networks. Thus a main contribution of this paper is showing that the first few eigenvectors of the geometric mean can still be computed efficiently. Our algorithm is based on the inverse power method and the extended Krylov subspace technique introduced by [8] and allows to compute eigenvectors of the geometric mean $A\#B$ of two matrices $A, B$ without ever computing $A\#B$ itself.

In Section 2 we discuss existing work on Laplacians on signed graphs. In Section 3 we discuss the geometric mean of two matrices and introduce the geometric mean Laplacian which is the basis of our spectral clustering method for signed graphs. In Section 4 we analyze our and existing approaches for the stochastic block model. In Section 5 we introduce our efficient algorithm to compute eigenvectors of the geometric mean of two matrices, and finally in Section 6 we discuss performance of our approach on real world graphs. *Proofs have been moved to the supplementary material.*

## 2 Signed graph clustering

Networks encoding positive and negative relations among the nodes can be represented by weighted signed graphs. Consider two symmetric non-negative weight matrices $W^+$ and $W^-$, a vertex set $V = \{v_1, \ldots, v_n\}$, and let $G^+ = (V, W^+)$ and $G^- = (V, W^-)$ be the induced graphs. A signed graph is the pair $G^\pm = (G^+, G^-)$ where $G^+$ and $G^-$ encode positive and the negative relations, respectively.

The concept of community in signed networks is typically related to the theory of social balance. This theory, as presented in [10, 3], is based on the analysis of affective ties, where positive ties are a source of balance whereas negative ties are considered as a source of imbalance in social groups.

**Definition 1** ([5], $k$-balance)**.** *A signed graph is $k$-balanced if the set of vertices can be partitioned into $k$ sets such that within the subsets there are only positive edges, and between them only negative.*

The presence of $k$-balance in $G^\pm$ implies the presence of $k$ groups of nodes being both assortative in $G^+$ and dissassortative in $G^-$. However this situation is fairly rare in real world networks and expecting communities in signed networks to be a perfectly balanced set of nodes is unrealistic.

In the next section we will show that Laplacians inspired by Definition 1 are based on some form of arithmetic mean of Laplacians. As an alternative we propose the geometric mean of Laplacians and show that it is able to recover communities when either $G^+$ is assortative, or $G^-$ is disassortative, or both. Results of this paper will make clear that the use of the geometric mean of Laplacians allows to recognize communities where previous approaches fail.

### 2.1 Laplacians on Unsigned Graphs

Spectral clustering of undirected, unsigned graphs using the Laplacian matrix is a well established technique (see [19] for an overview). Given an unsigned graph $G = (V, W)$, the Laplacian and its normalized version are defined as

$$L = D - W \qquad L_{\text{sym}} = D^{-1/2} L D^{-1/2} \qquad (1)$$

where $D_{ii} = \sum_{j=1}^n w_{ij}$ is the diagonal matrix of the degrees of $G$. Both Laplacians are positive semidefinite, and the multiplicity $k$ of the eigenvalue 0 is equal to the number of connected components in the graph. Further, the Laplacian is suitable in assortative cases [19], *i.e.* for the identification of clusters under the assumption that the amount of edges inside clusters has to be larger than the amount of edges between them.

For disassortative cases, *i.e.* for the identification of clusters where the amount of edges has to be larger between clusters than inside clusters, the signless Laplacian is a better choice [18]. Given the unsigned graph $G = (V, W)$, the signless Laplacian and its normalized version are defined as

$$Q = D + W, \qquad Q_{\text{sym}} = D^{-1/2} Q D^{-1/2} \qquad (2)$$

Both Laplacians are positive semi-definite, and the smallest eigenvalue is zero if and only if the graph has a bipartite component [6].

## 2.2 Laplacians on Signed Graphs

Recently a number of Laplacian operators for signed networks have been introduced. Consider the signed graph $G^\pm = (G^+, G^-)$. Let $D_{ii}^+ = \sum_{j=1}^n w_{ij}^+$ be the diagonal matrix of the degrees of $G^+$ and $\bar{D}_{ii} = \sum_{j=1}^n w_{ij}^+ + w_{ij}^-$ the one of the overall degrees in $G^\pm$.

The following Laplacians for signed networks have been considered so far

$$L_{BR} = D^+ - W^+ + W^-, \quad L_{BN} = \bar{D}^{-1} L_{BR}, \qquad \text{(balance ratio/normalized Laplacian)}$$
$$L_{SR} = \bar{D} - W^+ + W^-, \quad L_{SN} = \bar{D}^{-1/2} L_{SR} \bar{D}^{-1/2}, \quad \text{(signed ratio/normalized Laplacian)} \tag{3}$$

and spectral clustering algorithms have been proposed for $G^\pm$, based on these Laplacians [16, 4]. Let $L^+$ and $Q^-$ be the Laplacian and the signless Laplacian matrices of the graphs $G^+$ and $G^-$, respectively. We note that the matrix $L_{SR}$ blends the informations from $G^+$ and $G^-$ into (twice) the arithmetic mean of $L^+$ and $Q^-$, namely the following identity holds

$$L_{SR} = L^+ + Q^-. \tag{4}$$

Thus, as an alternative to the normalization defining $L_{SN}$ from $L_{SR}$, it is natural to consider the arithmetic mean of the normalized Laplacians $L_{AM} = L_{\text{sym}}^+ + Q_{\text{sym}}^-$. In the next section we introduce the geometric mean of $L_{\text{sym}}^+$ and $Q_{\text{sym}}^-$ and propose a new clustering algorithm for signed graphs based on that matrix. The analysis and experiments of next sections will show that blending the information from the positive and negative graphs trough the geometric mean overcomes the deficiencies showed by the arithmetic mean based operators.

# 3 Geometric mean of Laplacians

We define here the geometric mean of matrices and introduce the geometric mean of normalized Laplacians for clustering signed networks. Let $A^{1/2}$ be the unique positive definite solution of the matrix equation $X^2 = A$, where $A$ is positive definite.

**Definition 2.** *Let $A$, $B$ be positive definite matrices. The geometric mean of $A$ and $B$ is the positive definite matrix $A\#B$ defined by $A\#B = A^{1/2}(A^{-1/2}BA^{-1/2})^{1/2}A^{1/2}$.*

One can prove that $A\#B = B\#A$ (see [1] for details). Further, there are several useful ways to represent the geometric mean of positive definite matrices (see f.i. [1, 12])

$$A\#B = A(A^{-1}B)^{1/2} = (BA^{-1})^{1/2}A = B(B^{-1}A)^{1/2} = (AB^{-1})^{1/2}B \tag{5}$$

The next result reveals further consistency with the scalar case, in fact we observe that if $A$ and $B$ have some eigenvectors in common, then $A + B$ and $A\#B$ have those eigenvectors, with eigenvalues given by the arithmetic and geometric mean of the corresponding eigenvalues of $A$ and $B$, respectively.

**Theorem 1.** *Let $\mathbf{u}$ be an eigenvector of $A$ and $B$ with eigenvalues $\lambda$ and $\mu$, respectively. Then, $\mathbf{u}$ is an eigenvector of $A + B$ and $A\#B$ with eigenvalue $\lambda + \mu$ and $\sqrt{\lambda\mu}$, respectively.*

## 3.1 Geometric mean for signed networks clustering

Consider the signed network $G^\pm = (G^+, G^-)$. We define the normalized geometric mean Laplacian of $G^\pm$ as

$$L_{GM} = L_{\text{sym}}^+ \# Q_{\text{sym}}^- \tag{6}$$

We propose Algorithm 1 for clustering signed networks, based on the spectrum of $L_{GM}$. By definition 2, the matrix geometric mean $A\#B$ requires $A$ and $B$ to be positive definite. As both the Laplacian and the signless Laplacian are positve semi-definte, in what follows we shall assume that the matrices $L_{\text{sym}}^+$ and $Q_{\text{sym}}^-$ in (6) are modified by a small diagonal shift, ensuring positive definiteness. That is, in practice, we consider $L_{\text{sym}}^+ + \varepsilon_1 I$ and $Q_{\text{sym}}^- + \varepsilon_2 I$ being $\varepsilon_1$ and $\varepsilon_2$ small positive numbers. For the sake of brevity, we do not explicitly write the shifting matrices.

---

**Input**: Symmetric weight matrices $W^+, W^- \in \mathbb{R}^{n \times n}$, number $k$ of clusters to construct.
**Output**: Clusters $C_1, \dots, C_k$.
1 Compute the $k$ eigenvectors $\mathbf{u}_1, \dots, \mathbf{u}_k$ corresponding to the $k$ smallest eigenvalues of $L_{GM}$.
2 Let $U = (\mathbf{u}_1, \dots, \mathbf{u}_k)$.
3 Cluster the rows of $U$ with $k$-means into clusters $C_1, \dots, C_k$.
**Algorithm 1:** Spectral clustering with $L_{GM}$ on signed networks

| | | | |
|---|---|---|---|
| $(E_+)$ | $p_{\text{out}}^+ < p_{\text{in}}^+$ | $(E_{\text{vol}})$ | $p_{\text{in}}^- + (k-1)p_{\text{out}}^- < p_{\text{in}}^+ + (k-1)p_{\text{out}}^+$ |
| $(E_-)$ | $p_{\text{in}}^- < p_{\text{out}}^-$ | $(E_{\text{conf}})$ | $\left(\frac{kp_{\text{out}}^+}{p_{\text{in}}^+ + (k-1)p_{\text{out}}^+}\right)\left(\frac{kp_{\text{in}}^-}{p_{\text{in}}^- + (k-1)p_{\text{out}}^-}\right) < 1$ |
| $(E_{\text{bal}})$ | $p_{\text{in}}^- + p_{\text{out}}^+ < p_{\text{in}}^+ + p_{\text{out}}^-$ | $(E_G)$ | $\left(\frac{kp_{\text{out}}^+}{p_{\text{in}}^+ + (k-1)p_{\text{out}}^+}\right)\left(1 + \frac{p_{\text{in}}^- - p_{\text{out}}^-}{p_{\text{in}}^- + (k-1)p_{\text{out}}^-}\right) < 1$ |

Table 1: Conditions for the Stochastic Block Model analysis of Section 4

The main bottleneck of Algorithm 1 is the computation of the eigenvectors in step 1. In Section 5 we propose a scalable Krylov-based method to handle this problem.

Let us briefly discuss the motivating intuition behind the proposed clustering strategy. Algorithm 1, as well as state-of-the-art clustering algorithms based on the matrices in (3), rely on the $k$ smallest eigenvalues of the considered operator and their corresponding eigenvectors. Thus the relative ordering of the eigenvalues plays a crucial role. Assume the eigenvalues to be enumerated in ascending order. Theorem 1 states that the functions $(A, B) \mapsto A + B$ and $(A, B) \mapsto A \# B$ map eigenvalues of $A$ and $B$ having the same corresponding eigenvectors, into the arithmetic mean $\lambda_i(A) + \lambda_j(B)$ and geometric mean $\sqrt{\lambda_i(A)\lambda_j(B)}$, respectively, where $\lambda_i(\cdot)$ is the $i^{th}$ smallest eigenvalue of the corresponding matrix. Note that the indices $i$ and $j$ are not the same in general, as the eigenvectors shared by $A$ and $B$ may be associated to eigenvalues having different positions in the relative ordering of $A$ and $B$. This intuitively suggests that small eigenvalues of $A + B$ are related to small eigenvalues of both $A$ and $B$, whereas those of $A\#B$ are associated with small eigenvalues of either $A$ or $B$, or both. Therefore the relative ordering of the small eigenvalues of $L_{GM}$ is influenced by the presence of assortative clusters in $G^+$ (related to small eigenvalues of $L_{\text{sym}}^+$) or by disassortative clusters in $G^-$ (related to small eigenvalues in $Q_{\text{sym}}^-$), whereas the ordering of the small eigenvalues of the arithmetic mean takes into account only the presence of both those situations.

In the next section, for networks following the stochastic block model, we analyze in expectation the spectrum of the normalized geometric mean Laplacian as well as the one of the normalized Laplacians previously introduced. In this case the expected spectrum can be computed explicitly and we observe that in expectation the ordering induced by blending the informations of $G^+$ and $G^-$ trough the geometric mean allows to recover the ground truth clusters perfectly, whereas the use of the arithmetic mean introduces a bias which reverberates into a significantly higher clustering error.

## 4 Stochastic block model on signed graphs

In this section we present an analysis of different signed graph Laplacians based on the Stochastic Block Model (**SBM**). The SBM is a widespread benchmark generative model for networks showing a clustering, community, or group behaviour [22]. Given a prescribed set of groups of nodes, the SBM defines the presence of an edge as a random variable with probability being dependent on which groups it joins. To our knowledge this is the first analysis of spectral clustering on signed graphs with the stochastic block model. Let $\mathcal{C}_1, \ldots, \mathcal{C}_k$ be ground truth clusters, all having the same size $|\mathcal{C}|$. We let $p_{\text{in}}^+$ ($p_{\text{in}}^-$) be the probability that there exists a positive (negative) edge between nodes in the same cluster, and let $p_{\text{out}}^+$ ($p_{\text{out}}^-$) denote the probability of a positive (negative) edge between nodes in different clusters.

Calligraphic letters denote matrices in expectation. In particular $\mathcal{W}^+$ and $\mathcal{W}^-$ denote the weight matrices in expectation. We have $\mathcal{W}_{i,j}^+ = p_{\text{in}}^+$ and $\mathcal{W}_{i,j}^- = p_{\text{in}}^-$ if $v_i, v_j$ belong to the same cluster, whereas $\mathcal{W}_{i,j}^+ = p_{\text{out}}^+$ and $\mathcal{W}_{i,j}^- = p_{\text{out}}^-$ if $v_i, v_j$ belong to different clusters. Sorting nodes according to the ground truth clustering shows that $\mathcal{W}^+$ and $\mathcal{W}^-$ have rank $k$.

Consider the relations in Table 1. Conditions $E_+$ and $E_-$ describe the presence of assortative or disassortative clusters in expectation. Note that, by Definition 1, a graph is balanced if and only if $p_{\text{out}}^+ = p_{\text{in}}^- = 0$. We can see that if $E_+ \cap E_-$ then $G^-$ and $G^+$ give information about the cluster structure. Further, if $E_+ \cap E_-$ holds then $E_{\text{bal}}$ holds. Similarly $E_{\text{conf}}$ characterizes a graph where the relative amount of conflicts - *i.e.* positive edges between the clusters and negative edges inside the clusters - is small. Condition $E_G$ is strictly related to such setting. In fact when $E_- \cap E_G$ holds then

$E_{\text{conf}}$ holds. Finally condition $E_{\text{vol}}$ implies that the expected volume in the negative graph is smaller than the expected volume in the positive one. This condition is therefore not related to any signed clustering structure.

Let

$$\boldsymbol{\chi}_1 = \mathbf{1}, \qquad \boldsymbol{\chi}_i = (k-1)\mathbf{1}_{\mathcal{C}_i} - \mathbf{1}_{\overline{\mathcal{C}_i}}.$$

The use of $k$-means on $\boldsymbol{\chi}_i$, $i = 1, \dots, k$ identifies the ground truth communities $\mathcal{C}_i$. As spectral clustering relies on the eigenvectors corresponding to the $k$ smallest eigenvalues (see Algorithm 1) we derive here necessary and sufficient conditions such that in expectation the eigenvectors $\boldsymbol{\chi}_i$, $i = 1, \dots, k$ correspond to the $k$ smallest eigenvalues of the normalized Laplacians introduced so far. In particular, we observe that condition $E_G$ affects the ordering of the eigenvalues of the normalized geometric mean Laplacian. Instead, the ordering of the eigenvalues of the operators based on the arithmetic mean is related to $E_{\text{bal}}$ and $E_{\text{vol}}$. The latter is not related to any clustering, thus introduces a bias in the eigenvalues ordering which reverberates into a noisy embedding of the data points and in turn into a significantly higher clustering error.

**Theorem 2.** *Let $\mathcal{L}_{BN}$ and $\mathcal{L}_{SN}$ be the normalized Laplacians defined in (3) of the expected graphs. The following statements are equivalent:*

1.  *$\boldsymbol{\chi}_1, \dots, \boldsymbol{\chi}_k$ are the eigenvectors corresponding to the $k$ smallest eigenvalues of $\mathcal{L}_{BN}$.*
2.  *$\boldsymbol{\chi}_1, \dots, \boldsymbol{\chi}_k$ are the eigenvectors corresponding to the $k$ smallest eigenvalues of $\mathcal{L}_{SN}$.*
3.  *The two conditions $E_{\text{bal}}$ and $E_{\text{vol}}$ hold simultaneously.*

**Theorem 3.** *Let $\mathcal{L}_{GM} = \mathcal{L}_{\text{sym}}^{+}\#\mathcal{Q}_{\text{sym}}^{-}$ be the geometric mean of the Laplacians of the expected graphs. Then $\boldsymbol{\chi}_1, \dots, \boldsymbol{\chi}_k$ are the eigenvectors corresponding to the $k$ smallest eigenvalues of $\mathcal{L}_{GM}$ if and only if condition $E_G$ holds.*

Conditions for the geometric mean Laplacian of diagonally shifted Laplacians are available in the supplementary material. Intuition suggests that a good model should easily identify clusters when $E_+ \cap E_-$. However, unlike condition $E_G$, condition $E_{\text{vol}} \cap E_{\text{bal}}$ is not directly satisfied under that regime. Specifically, we have

**Corollary 1.** *Assume that $E_+ \cap E_-$ holds. Then $\boldsymbol{\chi}_1, \dots, \boldsymbol{\chi}_k$ are eigenvectors corresponding to the $k$ smallest eigenvalues of $\mathcal{L}_{GM}$. Let $p(k)$ denote the proportion of cases where $\boldsymbol{\chi}_1, \dots, \boldsymbol{\chi}_k$ are the eigenvectors of the $k$ smallest eigenvalues of $\mathcal{L}_{SN}$ or $\mathcal{L}_{BN}$, then $p(k) \leq \frac{1}{6} + \frac{2}{3(k-1)} + \frac{1}{(k-1)^2}$.*

In order to grasp the difference in expectation between $L_{BN}$, $L_{SN}$ and $L_{GM}$, in Fig 1 we present the proportion of cases where Theorems 2 and 3 hold under different contexts. Experiments are done with all four parameters discretized in $[0, 1]$ with 100 steps. The expected proportion of cases where $E_G$ holds (Theorem 3) is far above the corresponding proportion for $E_{\text{vol}} \cap E_{\text{bal}}$ (Theorem 2), showing that in expectation the geometric mean Laplacian is superior to the other signed Laplacians. In Fig. 2 we present experiments on sampled graphs with $k$-means on top of the $k$ smallest eigenvectors. In all cases we consider clusters of size $|\mathcal{C}| = 100$ and present the median of clustering error (i.e., error when clusters are labeled via majority vote) of 50 runs. The results show that the analysis made in expectation closely resembles the actual behavior. In fact, even if we expect only one noisy eigenvector for $L_{BN}$ and $L_{SN}$, the use of the geometric mean Laplacian significantly outperforms any other previously proposed technique in terms of clustering error. $L_{SN}$ and $L_{BN}$ achieve good clustering only when the graph resembles a $k$-balanced structure, whereas they fail even in the ideal situation where either the positive or the negative graphs are informative about the cluster structure. As shown in Section 6, the advantages of $L_{GM}$ over the other Laplacians discussed so far allow us to identify a clustering structure on the Wikipedia benchmark real world signed network, where other clustering approaches have failed.

# 5   Krylov-based inverse power method for small eigenvalues of $L_{\text{sym}}^{+}\#Q_{\text{sym}}^{-}$

The computation of the geometric mean $A\#B$ of two positive definite matrices of moderate size has been discussed extensively by various authors [20, 11, 12, 13]. However, when $A$ and $B$ have large dimensions, the approaches proposed so far become unfeasible, in fact $A\#B$ is in general a full matrix even if $A$ and $B$ are sparse. In this section we present a scalable algorithm for the computation of the smallest eigenvectors of $L_{\text{sym}}^{+}\#Q_{\text{sym}}^{-}$. The method is discussed for a general pair of matrices $A$ and $B$, to emphasize its general applicability which is therefore interesting in itself. We remark that

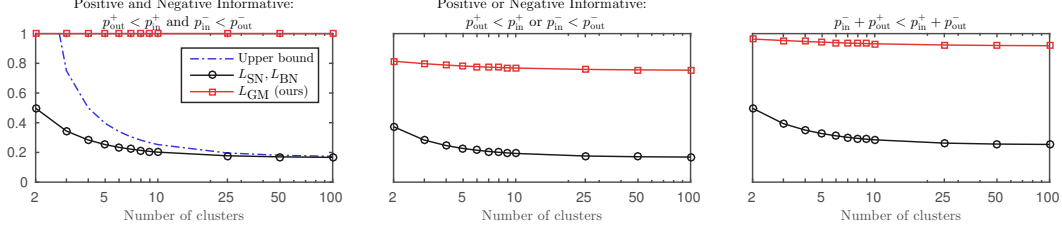

Figure 1: Fraction of cases where in expectation $\boldsymbol{\chi}_1, \ldots, \boldsymbol{\chi}_k$ correspond to the $k$ smallest eigenvalues under the SBM.

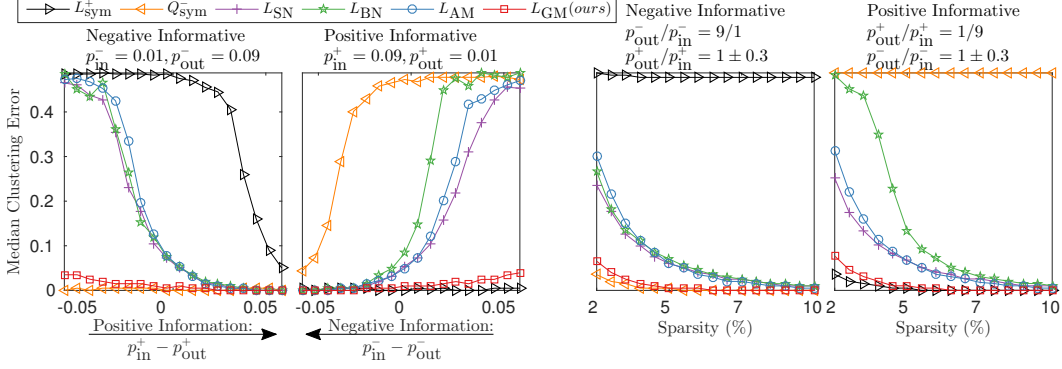

Figure 2: Median clustering error under the stochastic block model over 50 runs.

the method takes advantage of the sparsity of $A$ and $B$ and does not require to explicitly compute the matrix $A\#B$. To our knowledge this is the first effective method explicitly built for the computation of the eigenvectors of the geometric mean of two large and sparse positive definite matrices.

Given a positive definite matrix $M$ with eigenvalues $\lambda_1 \leq \cdots \leq \lambda_n$, let $\mathcal{H}$ be any eigenspace of $M$ associated to $\lambda_1, \ldots, \lambda_t$. The inverse power method (**IPM**) applied to $M$ is a method that converges to an eigenvector $\mathbf{x}$ associated to the smallest eigenvalue $\lambda_{\mathcal{H}}$ of $M$ such that $\lambda_{\mathcal{H}} \neq \lambda_i, i = 1, \ldots, t$. The pseudocode of IPM applied to $A\#B = A(A^{-1}B)^{1/2}$ is shown in Algorithm 2. Given a vector $\mathbf{v}$ and a matrix $M$, the notation $\mathrm{solve}\{M, \mathbf{v}\}$ is used to denote a procedure returning the solution $\mathbf{x}$ of the linear system $M\mathbf{x} = \mathbf{v}$. At each step the algorithm requires the solution of two linear systems. The first one (line 2) is solved by the preconditioned conjugate gradient method, where the preconditioner is obtained by the incomplete Cholesky decomposition of $A$. Note that the conjugate gradient method is very fast, as $A$ is assumed sparse and positive definite, and it is matrix-free, *i.e.* it requires to compute the action of $A$ on a vector, whereas it does not require the knowledge of $A$ (nor its inverse). The solution of the linear system occurring in line 3 is the major inner-problem of the proposed algorithm. Its efficient solution is performed by means of an extended Krylov subspace technique that we describe in the next section. The proposed implementation ensures the whole IPM is matrix-free and scalable.

## 5.1 Extended Krylov subspace method for the solution of the linear system $(A^{-1}B)^{1/2}\mathbf{x} = \mathbf{y}$

We discuss here how to apply the technique known as Extended Krylov Subspace Method (**EKSM**) for the solution of the linear system $(A^{-1}B)^{1/2}\mathbf{x} = \mathbf{y}$. Let $M$ be a large and sparse matrix, and $\mathbf{y}$ a given vector. When $f$ is a function with a single pole, EKSM is a very effective method to approximate the vector $f(M)\mathbf{y}$ without ever computing the matrix $f(M)$ [8]. Note that, given two positive definite matrices $A$ and $B$ and a vector $\mathbf{y}$, the vector we want to compute is $\mathbf{x} = (A^{-1}B)^{-1/2}\mathbf{y}$, so that our problem boils down to the computation of the product $f(M)\mathbf{y}$, where $M = A^{-1}B$ and $f(X) = X^{-1/2}$. The general idea of EKSM $s$-th iteration is to project $M$ onto the subspace

$$\mathbb{K}^s(M, \mathbf{y}) = \mathrm{span}\{\mathbf{y}, M\mathbf{y}, M^{-1}\mathbf{y}, \ldots, M^{s-1}\mathbf{y}, M^{1-s}\mathbf{y}\},$$

and solve the problem there. The projection onto $\mathbb{K}^s(M, \mathbf{y})$ is realized by means of the Lanczos process, which produces a sequence of matrices $V_s$ with orthogonal columns, such that the first

column of $V_s$ is a multiple of $\mathbf{y}$ and $\mathrm{range}(V_s) = \mathbb{K}^s(M, \mathbf{y})$. Moreover at each step we have

$$MV_s = V_sH_s + [\mathbf{u}_{s+1}, \mathbf{v}_{s+1}][\mathbf{e}_{2s+1}, \mathbf{e}_{2s+2}]^T \qquad (7)$$

where $H_s$ is $2s \times 2s$ symmetric tridiagonal, $\mathbf{u}_{s+1}$ and $\mathbf{v}_{s+1}$ are orthogonal to $V_s$, and $\mathbf{e}_i$ is the $i$-th canonical vector. The solution $\mathbf{x}$ is then approximated by $\mathbf{x}_s = V_s f(H_s)\mathbf{e}_1 \|\mathbf{y}\| \approx f(M)\mathbf{y}$. If $n$ is the order of $M$, then the exact solution is obtained after at most $n$ steps. However, in practice, significantly fewer iterations are enough to achieve a good approximation, as the error $\|\mathbf{x}_s - \mathbf{x}\|$ decays exponentially with $s$ (Thm 3.4 and Prop. 3.6 in [14]). See the supplementary material for details.

The pseudocode for the extended Krylov iteration is presented in Algorithm 3. We use the stopping criterion proposed in [14]. It is worth pointing out that at step 4 of the algorithm we can freely choose any scalar product $\langle \cdot, \cdot \rangle$, without affecting formula (7) nor the convergence properties of the method. As $M = A^{-1}B$, we use the scalar product $\langle \mathbf{u}, \mathbf{v} \rangle_A = \mathbf{u}^T A\mathbf{v}$ induced by the positive definite matrix $A$, so that the computation of the tridiagonal matrix $H_s$ in the algorithm simplifies to $V_s^T BV_s$. We refer to [9] for further details. As before, the solve procedure is implemented by means of the preconditioned conjugate gradient method, where the preconditioner is obtained by the incomplete Cholesky decomposition of the coefficient matrix. Figure 3 shows that we are able to compute the smallest eigenvector of $L_{\mathrm{sym}}^+ \# Q_{\mathrm{sym}}^-$ being just a constant factor worse than the computation of the eigenvector of the arithmetic mean, whereas the direct computation of the geometric mean followed by the computation of the eigenvectors is unfeasible for large graphs.

---

**Input**: $\mathbf{x}_0$, eigenspace $\mathcal{H}$ of $A\#B$.
**Output**: Eigenpair $(\lambda_{\mathcal{H}}, \mathbf{x})$ of $A\#B$
1  **repeat**
2     $\mathbf{u}_k \leftarrow \mathrm{solve}\{A, \mathbf{x}_k\}$
3     $\mathbf{v}_k \leftarrow \mathrm{solve}\{(A^{-1}B)^{1/2}, \mathbf{u}_k\}$
4     $\mathbf{y}_k \leftarrow$ project $\mathbf{u}_k$ over $\mathcal{H}^{\perp}$
5     $\mathbf{x}_{k+1} \leftarrow \mathbf{y}_k / \|\mathbf{y}_k\|_2$
6  **until** *tolerance reached*
7  $\lambda_{\mathcal{H}} \leftarrow \mathbf{x}_{k+1}^T\mathbf{x}_k, \quad \mathbf{x} \leftarrow \mathbf{x}_{k+1}$
**Algorithm 2**: IPM applied to $A\#B$.

**Input**: $\mathbf{u}_0 = \mathbf{y}, V_0 = [\cdot]$
**Output**: $\mathbf{x} = (A^{-1}B)^{-1/2}\mathbf{y}$
1  $\mathbf{v}_0 \leftarrow \mathrm{solve}\{B, A\mathbf{u}_0\}$
2  **for** $s = 0, 1, 2, \ldots, n$ **do**
3     $\tilde{V}_{s+1} \leftarrow [V_s, \mathbf{u}_s, \mathbf{v}_s]$
4     $V_{s+1} \leftarrow$ Orthogonalize columns of $\tilde{V}_{s+1}$ w.r.t. $\langle \cdot, \cdot \rangle_A$
5     $H_{s+1} \leftarrow V_{s+1}^T BV_{s+1}$
6     $\mathbf{x}_{s+1} \leftarrow H_{s+1}^{-1/2}\mathbf{e}_1$
7     **if** *tolerance reached* **then** *break*
8     $\mathbf{u}_{s+1} \leftarrow \mathrm{solve}\{A, BV_{s+1}\mathbf{e}_1\}$
9     $\mathbf{v}_{s+1} \leftarrow \mathrm{solve}\{B, AV_{s+1}\mathbf{e}_2\}$
10 **end**
11 $\mathbf{x} \leftarrow V_{s+1}\mathbf{x}_{s+1}$
**Algorithm 3**: EKSM for the computation of $(A^{-1}B)^{-1/2}\mathbf{y}$

---

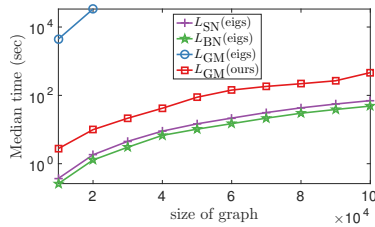

Figure 3: Median execution time of 10 runs for different Laplacians. Graphs have two perfect clusters and $2.5\%$ of edges among nodes. $L_{GM}(ours)$ uses Algs 2 and 3, whereas we used Matlab's eigs for the other matrices. The use of eigs on $L_{GM}$ is prohibitive as it needs the matrix $L_{GM}$ to be built (we use the toolbox provided in [2]), destroying the sparsity of the original graphs. Experiments are performed using one thread.

# 6 Experiments

**Sociology Networks** We evaluate signed Laplacians $L_{SN}, L_{BN}, L_{AM}$ and $L_{GM}$ through three real-world and moderate size signed networks: Highland tribes (Gahuku-Gama) network [21], Slovene Parliamentary Parties Network [15] and US Supreme Court Justices Network [7]. For the sake of comparison we take as ground truth the clustering that is stated in the corresponding references. We observe that all signed Laplacians yield zero clustering error.

**Experiments on Wikipedia signed network.** We consider the Wikipedia adminship election dataset from [17], which describes relationships that are positive, negative or non existent. We use Algs. $1-3$ and look for 30 clusters. Positive and negative adjacency matrices sorted according to our clustering are depicted in Figs. 4(a) and 4(b). We can observe the presence of a large relatively empty cluster.

Zooming into the denser portion of the graph we can see a $k$-balanced behavior (see Figs. 4(c) and 4(d)), *i.e.* the positive adjacency matrix shows assortative groups - resembling a block diagonal structure - while the negative adjacency matrix shows a disassortative setting. Using $L_{AM}$ and $L_{BN}$ we were not able to find any clustering structure, which corroborates results reported in [4]. This further confirms that $L_{GM}$ overcomes other clustering approaches. To the knowledge of the authors, this is the first time that clustering structure has been found in this dataset.

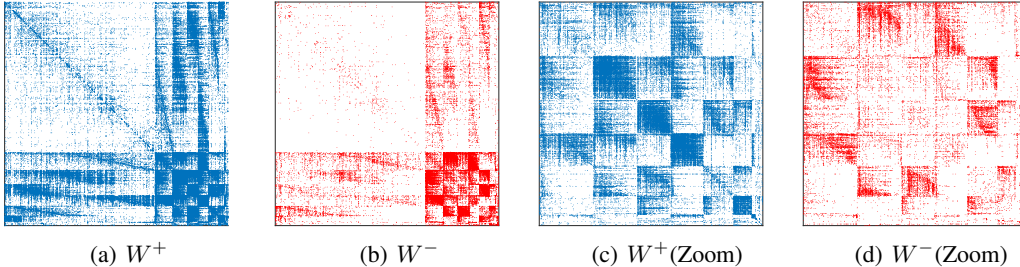

| (a) $W^+$ | (b) $W^-$ | (c) $W^+$(Zoom) | (d) $W^-$(Zoom) |

Figure 4: Wikipedia weight matrices sorted according to the clustering obtained with $L_{GM}$ (Alg. 1).

**Experiments on UCI datasets.** We evaluate our method $L_{GM}$ (Algs. 1−3) against $L_{SN}$, $L_{BN}$, and $L_{AM}$ with datasets from the UCI repository (see Table. 2). We build $W^+$ from a symmetric $k^+$-nearest neighbor graph, whereas $W^-$ is obtained from the symmetric $k^-$-farthest neighbor graph. For each dataset we test all clustering methods over all possible choices of $k^+, k^- \in \{3, 5, 7, 10, 15, 20, 40, 60\}$. In Table 2 we report the fraction of cases where each method achieves the best and strictly best clustering error over all the 64 graphs, per each dataset. We can see that our method outperforms other methods across all datasets.

In the figure on the right of Table 2 we present the clustering error on MNIST dataset fixing $k^+ = 10$. With $Q_{\text{sym}}^-$ one gets the highest clustering error, which shows that the $k^-$-farthest neighbor graph is a source of noise and is not informative. In fact, we observe that a small subset of nodes is the farthest neighborhood of a large fraction of nodes. The noise from the $k^-$-farthest neighbor graph is strongly influencing the performances of $L_{SN}$ and $L_{BN}$, leading to a noisy embedding of the datapoints and in turn to a high clustering error. On the other hand we can see that $L_{GM}$ is robust, in the sense that its clustering performances are not affected negatively by the noise in the negative edges. Similar behaviors have been observed for the other datasets in Table 2, and are shown in supplementary material.

| | | iris | wine | ecoli | optdig | USPS | pendig | MNIST |
|---|---|---|---|---|---|---|---|---|
| # vertices | | 150 | 178 | 310 | 5620 | 9298 | 10992 | 70000 |
| # classes | | 3 | 3 | 3 | 10 | 10 | 10 | 10 |
| $L_{SN}$ | Best (%) | 23.4 | 40.6 | 18.8 | 28.1 | 10.9 | 10.9 | 12.5 |
| | Str. best (%) | 10.9 | 21.9 | 14.1 | 28.1 | 9.4 | 10.9 | 12.5 |
| $L_{BN}$ | Best (%) | 17.2 | 21.9 | 7.8 | 0.0 | 1.6 | 3.1 | 0.0 |
| | Str. best (%) | 7.8 | 4.7 | 6.3 | 0.0 | 1.6 | 3.1 | 0.0 |
| $L_{AM}$ | Best (%) | 12.5 | 28.1 | 14.1 | 0.0 | 0.0 | 1.6 | 0.0 |
| | Str. best (%) | 10.9 | 14.1 | 12.5 | 0.0 | 0.0 | 1.6 | 0.0 |
| $\boldsymbol{L_{GM}}$ | Best (%) | **59.4** | **42.2** | **65.6** | **71.9** | **89.1** | **84.4** | **87.5** |
| | Str. best (%) | **57.8** | **35.9** | **60.9** | **71.9** | **87.5** | **84.4** | **87.5** |

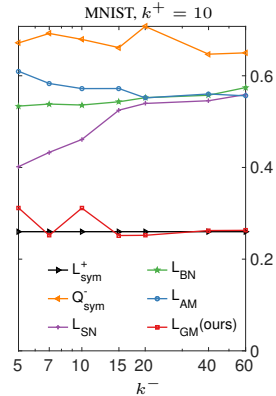

Table 2: Experiments on UCI datasets. Left: fraction of cases where methods achieve best and strictly best clustering error. Right: clustering error on MNIST dataset.

**Acknowledgments.** The authors acknowledge support by the ERC starting grant NOLEPRO

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
