[Supplementary Material · camera_ready_long_submitted_spaceCareV2.pdf]

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

*Proof.* Using the identities $A\mathbf{u} = \lambda\mathbf{u}$ and $B\mathbf{u} = \mu\mathbf{u}$ we have $(A + B)\mathbf{u} = (\lambda + \mu)\mathbf{u}$. For the geometric mean, observe that for any positive definite matrix $M$, if $M\mathbf{x} = \lambda(M)\mathbf{x}$, then $M^{1/2}\mathbf{x} = \lambda(M)^{1/2}\mathbf{x}$. In particular we have

$$
A^{-1/2} B A^{-1/2} \mathbf{u} = \lambda^{-1/2} A^{-1/2} B \mathbf{u} = \lambda^{-1/2} \mu A^{-1/2} \mathbf{u} = (\mu/\lambda)\mathbf{u}
$$

thus $(A^{-1/2} B A^{-1/2})^{1/2}\mathbf{u} = \sqrt{\mu/\lambda}\,\mathbf{u}$. As a consequence

$$
(A \# B)\mathbf{u} = A^{1/2}(A^{-1/2} B A^{-1/2})^{1/2} A^{1/2}\mathbf{u} = \lambda^{1/2} A^{1/2}(A^{-1/2} B A^{-1/2})^{1/2}\mathbf{u} = (\sqrt{\lambda\mu})\mathbf{u}
$$

which concludes the proof. $\square$

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

. It is simple to verify that $\boldsymbol{\chi}_i$ are eigenvectors of $\mathcal{W}^+$ and $\mathcal{W}^-$, with eigenvalues denoted by $\lambda_i^+$ and $\lambda_i^-$, respectively. Thus $\mathcal{W}^+$ and $\mathcal{W}^-$ are simultaneously diagonalizable, that is there exists a non-singular matrix $\Sigma$ such that $\Sigma^{-1}\mathcal{W}^\pm\Sigma = \Lambda^\pm$, where $\Lambda^+$ and $\Lambda^-$ are diagonal matrices $\Lambda^\pm = \text{diag}(\lambda_1^\pm, \ldots, \lambda_k^\pm, 0, \ldots, 0)$. Observe that the eigenvalues $\lambda_i^+$ and $\lambda_i^-$ admits the following explicit representations

$$
\begin{aligned}
\lambda_1^+ &= |\mathcal{C}|\,(p_{\text{in}}^+ + (k-1)p_{\text{out}}^+), & \lambda_1^- &= |\mathcal{C}|\,(p_{\text{in}}^- + (k-1)p_{\text{out}}^-) \\
\lambda_i^+ &= |\mathcal{C}|\,(p_{\text{in}}^+ - p_{\text{out}}^+) & \lambda_i^- &= |\mathcal{C}|\,(p_{\text{in}}^- - p_{\text{out}}^-),
\end{aligned}
\tag{7}
$$

for $i = 2, \ldots, k$. As we assume clusters of the same size, the nodes have the same degree in expectation, inducing a regular graph. Hence the expected degrees of the graph are $d^+ = \mathcal{W}^+\mathbf{1} = |\mathcal{C}|\,(p_{\text{in}}^+ + (k-1)p_{\text{out}}^+)\mathbf{1}$, $d^- = \mathcal{W}^-\mathbf{1} = |\mathcal{C}|\,(p_{\text{in}}^- + (k-1)p_{\text{out}}^-)\mathbf{1}$ and $\bar{d} = d^+ + d^-$. With corresponding degree matrices $\mathcal{D}^+ = d^+ I$ and $\bar{\mathcal{D}} = \bar{d}I$. The expected balanced-ratio cut Laplacian operator is thus given by $\mathcal{L}_{BR} = \Sigma(d^+ I - \Lambda^+ + \Lambda^-)\Sigma^{-1}$. It follows that the eigenvalues of $\mathcal{L}_{BR}$ correspond to eigenvectors in the following way

$$
\begin{cases}
d^+ - \lambda_i^+ + \lambda_i^- & \text{with eigenvector } \boldsymbol{\chi}_i, i = 1, \ldots, k \\
d^+ & \text{corresponding to the remaining eigenvectors}
\end{cases}
$$

Thus, eigenvectors $\boldsymbol{\chi}_i, i = 1, \ldots, k$ correspond to the smallest eigenvalues if and only if

$$d^+ - \lambda_i^+ + \lambda_i^- < d^+ \iff \lambda_i^- < \lambda_i^+$$

By Eqs. (7) we see that for the constant eigenvector we have

$$\lambda_1^- < \lambda_1^+ \iff d^- < d^+ \iff p_{\text{in}}^- + (k-1)p_{\text{out}}^- < p_{\text{in}}^+ + (k-1)p_{\text{out}}^+ \,,$$

whereas for the eigenvectors $\chi_i, i = 2, \ldots, k$ the corresponding condition is

$$\lambda_i^- < \lambda_i^+ \iff p_{\text{in}}^- + p_{\text{out}}^+ < p_{\text{in}}^+ + p_{\text{out}}^- \,.$$

We deduce that the eigenvectors $\chi_i, i = 1, \ldots, k$ correspond to the smallest eigenvalues of $\mathcal{L}_{BR}$ if and only if $p_{\text{in}}^- + (k-1)p_{\text{out}}^- < p_{\text{in}}^+ + (k-1)p_{\text{out}}^+$ and $p_{\text{in}}^- + p_{\text{out}}^+ < p_{\text{in}}^+ + p_{\text{out}}^-$.

As $\mathcal{L}_{BN}$ differs from $\mathcal{L}_{BR}$ by a constant factor, the conditions hold for $\mathcal{L}_{BN}$. Conditions for $\mathcal{L}_{SN}$ can be proved in the same way, as the only difference in the eigenvalues is a shift given by the degree vector $d$. □

**Theorem 3.** *Let $\mathcal{L}_{GM} = \mathcal{L}_{\text{sym}}^+ \# \mathcal{Q}_{\text{sym}}^-$ be the geometric mean of the Laplacians of the expected graphs. Then $\chi_1, \ldots, \chi_k$ are the eigenvectors corresponding to the $k$ smallest eigenvalues of $\mathcal{L}_{GM}$ if and only if condition $E_G$ holds.*

*Proof.* We use the same notation as in the proof of Theorem 2. Observing that $\mathcal{L}_{\text{sym}}^+$ and $\mathcal{Q}_{\text{sym}}^-$ have the same eigenvectors, it follows from Theorem 1 that

$$\mathcal{L}_{GM} = \Sigma \sqrt{(I - \widehat{\Lambda}^+)(I + \widehat{\Lambda}^-)} \, \Sigma^{-1} \tag{8}$$

where $d^+ \widehat{\Lambda}^+ = \Lambda^+$, and $d^- \widehat{\Lambda}^- = \Lambda^-$. We deduce that the eigenvalues of $\mathcal{L}_{GM}$ correspond to eigenvectors in the following way

$$\begin{cases} \sqrt{\left(1 - \frac{\lambda_i^+}{d^+}\right)\left(1 + \frac{\lambda_i^-}{d^-}\right)} & \text{with eigenvector } \chi_i, i = 1, \ldots, k \\ 1 & \text{corresponding to the remaining eigenvectors} \end{cases}$$

Thus, eigenvectors $\chi_i, i = 1, \ldots, k$ correspond to the smallest eigenvalues if and only if

$$\left(1 - \frac{\lambda_i^+}{d^+}\right)\left(1 + \frac{\lambda_i^-}{d^-}\right) < 1$$

By eqs. (7) we see that for the constant eigenvector $\chi_1$ we have

$$\left(1 - \frac{\lambda_1^+}{d^+}\right)\left(1 + \frac{\lambda_1^-}{d^-}\right) = \left(1 - \frac{d^+}{d^+}\right)\left(1 + \frac{d^-}{d^-}\right) = 0 < 1 \,.$$

For eigenvectors $\chi_i, i = 2, \ldots, k$ first observe that

$$1 - \frac{\lambda_i^+}{d^+} = (d^+ - \lambda_i^+)/d^+ = \left(d^+ - |\mathcal{C}|\left(p_{\text{in}}^+ - p_{\text{out}}^+\right)\right)/d^+ = \frac{k p_{\text{out}}^+}{p_{\text{in}}^+ + (k-1)p_{\text{out}}^+}$$

In the same way we have

$$1 + \frac{\lambda_i^-}{d^-} = 1 + \frac{p_{\text{in}}^- - p_{\text{out}}^-}{p_{\text{in}}^- + (k-1)p_{\text{out}}^-} \,.$$

Thus, for the eigenvectors $\chi_i, i = 2, \ldots, k$ we have the following condition

$$\left(1 - \frac{\lambda_i^+}{d^+}\right)\left(1 + \frac{\lambda_i^-}{d^-}\right) < 1 \iff \left(\frac{k p_{\text{out}}^+}{p_{\text{in}}^+ + (k-1)p_{\text{out}}^+}\right)\left(1 + \frac{p_{\text{in}}^- - p_{\text{out}}^-}{p_{\text{in}}^- + (k-1)p_{\text{out}}^-}\right) < 1 \,,$$

which implies in turn that the eigenvectors $\chi_i, i = 1, \ldots, k$ correspond to the smallest eigenvalues of $\mathcal{L}_{GM}$ if and only if $E_G$ holds. □

As mentioned above, in practical implementations one modifies the Laplacians defining $L_{GM}$ by adding a small diagonal shift. This is done to ensure positive definiteness of the matrices. The next theorem shows how to extend the previous result to the case of diagonally shifted Laplacians.

**Theorem 4.** *Let $\mathcal{L}_{GM} = (\mathcal{L}^+_{\text{sym}} + \varepsilon_1 I)\#(\mathcal{Q}^-_{\text{sym}} + \varepsilon_2 I)$ be the geometric mean of the shifted Laplacians of the expected graphs. Then $\chi_1, \ldots, \chi_k$ are the eigenvectors corresponding to the $k$ smallest eigenvalues of $\mathcal{L}_{GM}$ if the following conditions hold.*

&emsp; *1. $\varepsilon_1 + \varepsilon_2 < 1$.*

&emsp; *2. $\left(\dfrac{kp^+_{\text{out}}}{p^+_{\text{in}}+(k-1)p^+_{\text{out}}}\right)\left(1 + \dfrac{p^-_{\text{in}}-p^-_{\text{out}}}{p^-_{\text{in}}+(k-1)p^-_{\text{out}}}\right) + (\varepsilon_1 + \varepsilon_2) < 1.$*

*Proof.* We use the same notation as in the previous proof. Observing that $\mathcal{L}^+_{\text{sym}}$ and $\mathcal{Q}^-_{\text{sym}}$ have the same eigenvectors, it follows from Theorem 1 that

$$\mathcal{L}_{GM} = \Sigma \sqrt{(I - \widehat{\Lambda}^+ + \varepsilon_1 I)(I + \widehat{\Lambda}^- + \varepsilon_2 I)}\, \Sigma^{-1} \tag{9}$$

where $d^+\widehat{\Lambda}^+ = \Lambda^+$, and $d^-\widehat{\Lambda}^- = \Lambda^-$. We deduce that the eigenvalues of $\mathcal{L}_{GM}$ correspond to eigenvectors in the following way

$$\begin{cases} \sqrt{\left(1 - \frac{\lambda^+_i}{d^+} + \varepsilon_1\right)\left(1 + \frac{\lambda^-_i}{d^-} + \varepsilon_2\right)} & \text{with eigenvector } \chi_i, i = 1, \ldots, k \\ (1 + \varepsilon_1)(1 + \varepsilon_2) & \text{corresponding to the remaining eigenvectors} \end{cases}$$

Thus, eigenvectors $\chi_i, i = 1, \ldots, k$ correspond to the smallest eigenvalues if and only if

$$\left(1 - \frac{\lambda^+_i}{d^+} + \varepsilon_1\right)\left(1 + \frac{\lambda^-_i}{d^-} + \varepsilon_2\right) < (1 + \varepsilon_1)(1 + \varepsilon_2) \tag{10}$$

Further, we can see that the previous equation holds if and only if

$$\left(1 - \frac{\lambda^+_i}{d^+}\right)\left(1 + \frac{\lambda^-_i}{d^-}\right) + \varepsilon_1\left(1 + \frac{\lambda^-_i}{d^-}\right) + \varepsilon_2\left(1 - \frac{\lambda^+_i}{d^+}\right) < 1 + \varepsilon_1 + \varepsilon_2$$

More over, as $\left(1 - \frac{\lambda^+_i}{d^+}\right), \left(1 + \frac{\lambda^-_i}{d^-}\right) \in [0, 2]$, we can see that eq.(10) holds if

$$\left(1 - \frac{\lambda^+_i}{d^+}\right)\left(1 + \frac{\lambda^-_i}{d^-}\right) + \varepsilon_1 + \varepsilon_2 < 1$$

By eqs. (7) we see that for the constant eigenvector $\chi_1$ we have $1 - \frac{\lambda^+_1}{d^+} = 1 - \frac{d^+}{d^+} = 0$. Thus,

$$\left(1 - \frac{\lambda^+_i}{d^+}\right)\left(1 + \frac{\lambda^-_i}{d^-}\right) + \varepsilon_1 + \varepsilon_2 = \varepsilon_1 + \varepsilon_2 < 1$$

For eigenvectors $\chi_i, i = 2, \ldots, k$ first observe that

$$1 - \frac{\lambda^+_i}{d^+} = (d^+ - \lambda^+_i)/d^+ = \left(d^+ - |\mathcal{C}|\,(p^+_{\text{in}} - p^+_{\text{out}})\right)/d^+ = \frac{kp^+_{\text{out}}}{p^+_{\text{in}} + (k-1)p^+_{\text{out}}}$$

In the same way we have

$$1 + \frac{\lambda^-_i}{d^-} = 1 + \frac{p^-_{\text{in}} - p^-_{\text{out}}}{p^-_{\text{in}} + (k-1)p^-_{\text{out}}}\, .$$

Thus, for the eigenvectors $\chi_i, i = 2, \ldots, k$ we have the following condition

$$\left(1 - \frac{\lambda^+_i}{d^+}\right)\left(1 + \frac{\lambda^-_i}{d^-}\right) + \varepsilon_1 + \varepsilon_2 < 1 \iff$$

$$\left(\frac{kp^+_{\text{out}}}{p^+_{\text{in}} + (k-1)p^+_{\text{out}}}\right)\left(1 + \frac{p^-_{\text{in}} - p^-_{\text{out}}}{p^-_{\text{in}} + (k-1)p^-_{\text{out}}}\right) + \varepsilon_1 + \varepsilon_2 < 1,$$

This implies in turn that the eigenvectors $\chi_i, i = 1, \ldots, k$ correspond to the smallest eigenvalues of $\mathcal{L}_{GM}$ if the following conditions hold

&emsp; 1. $\varepsilon_1 + \varepsilon_2 < 1$.

&emsp; 2. $\left(\dfrac{kp^+_{\text{out}}}{p^+_{\text{in}}+(k-1)p^+_{\text{out}}}\right)\left(1 + \dfrac{p^-_{\text{in}}-p^-_{\text{out}}}{p^-_{\text{in}}+(k-1)p^-_{\text{out}}}\right) + (\varepsilon_1 + \varepsilon_2) < 1.$

<div style="text-align:right">□</div>

Intuition suggests that a good model should easily identify clusters when $E_+ \cap E_-$. However, unlike condition $E_G$, condition $E_{\text{vol}} \cap E_{\text{bal}}$ is not directly satisfied under that regime. Specifically, we have

**Corollary 1.** *Assume that $E_+ \cap E_-$ holds. Then $\chi_1, \ldots, \chi_k$ are eigenvectors corresponding to the $k$ smallest eigenvalues of $\mathcal{L}_{GM}$. Let $p(k)$ denote the proportion of cases where $\chi_1, \ldots, \chi_k$ are the eigenvectors of the $k$ smallest eigenvalues of $\mathcal{L}_{SN}$ or $\mathcal{L}_{BN}$, then $p(k) \leq \frac{1}{6} + \frac{2}{3(k-1)} + \frac{1}{(k-1)^2}$.*

*Proof.* The event $E_{\text{vol}}$ is defined as

$$E_{\text{vol}} = \{(p_{\text{in}}^-, p_{\text{out}}^-, p_{\text{in}}^+, p_{\text{out}}^+) \in [0,1]^4 \,|\, p_{\text{in}}^- + (k-1)p_{\text{out}}^- < p_{\text{in}}^+ + (k-1)p_{\text{out}}^+\}$$

We can rewrite the inequality as

$$p_{\text{out}}^- - p_{\text{out}}^+ < \frac{1}{k-1}\left(p_{\text{in}}^+ - p_{\text{in}}^-\right) < \frac{1}{k-1}.$$

Thus the event $\tilde{E}_{\text{vol}}$ defined as

$$\tilde{E}_{\text{vol}} = \{(p_{\text{in}}^-, p_{\text{out}}^-, p_{\text{in}}^+, p_{\text{out}}^+) \in [0,1]^4 \,|\, p_{\text{out}}^- - p_{\text{out}}^+ < \frac{1}{k-1}\},$$

satisfies $E_{\text{vol}} \subset \tilde{E}_{\text{vol}}$. Then with

$$E_3 = E_+ \cap E_- = \{(p_{\text{in}}^-, p_{\text{out}}^-) \in [0,1]^2 \,|\, p_{\text{in}}^- < p_{\text{out}}^-\} \cap \{(p_{\text{in}}^+, p_{\text{out}}^+) \in [0,1]^2 \,|\, p_{\text{out}}^+ < p_{\text{in}}^+\}$$
$$E_B = \{(p_{\text{in}}^-, p_{\text{out}}^-, p_{\text{in}}^+, p_{\text{out}}^+) \in [0,1]^4 \,|\, p_{\text{in}}^- + p_{\text{out}}^+ < p_{\text{in}}^+ + p_{\text{out}}^-\}$$

we observe $E_3 \subset E_B$. Then

$$p(k) = \mathrm{P}(E_B \cap E_{\text{vol}} \,|\, E_3) = \frac{\mathrm{P}(E_B \cap E_{\text{vol}} \cap E_3)}{\mathrm{P}(E_3)} = \frac{\mathrm{P}(E_{\text{vol}} \cap E_3)}{\mathrm{P}(E_3)}$$

$$\leq \frac{\mathrm{P}(\tilde{E}_{\text{vol}} \cap E_3)}{\mathrm{P}(E_3)}$$

Then we get with $(x_1, x_2, x_3, x_4)$ corresponding to $(p_{\text{in}}^+, p_{\text{out}}^+, p_{\text{out}}^-, p_{\text{in}}^-)$

$$\mathrm{P}(\tilde{E}_{\text{vol}} \cap E_3) \leq \int_0^1 \left(\int_0^{x_1}\left(\int_0^{x_2+\frac{1}{k-1}}\left(\int_0^{x_3} dx_4\right)dx_3\right)dx_2\right)dx_1$$

$$= \int_0^1 \left(\int_0^{x_1}\left(\int_0^{x_2+\frac{1}{k-1}} x_3 dx_3\right)dx_2\right)dx_1$$

$$= \int_0^1 \left(\int_0^{x_1}\left(\frac{1}{2}\left(x_2+\frac{1}{k-1}\right)^2\right)dx_2\right)dx_1$$

$$= \int_0^1 \left[\frac{1}{6}\left(x_1+\frac{1}{k-1}\right)^3\right]_0^{x_1} dx_1$$

$$= \int_0^1 \left[\frac{x_1^3}{6} + \frac{x_1^2}{2(k-1)} + \frac{x_1}{2(k-1)^2}\right]dx_1$$

$$= \left[\frac{x_1^4}{24} + \frac{x_1^3}{6(k-1)} + \frac{x_1^2}{4(k-1)^2}\right]_0^1$$

$$= \frac{1}{24} + \frac{1}{6(k-1)} + \frac{1}{4(k-1)^2}$$

The first inequality comes from the fact that we do not ensure that the integration upper border for $x_3$ is smaller or equal to one. Thus with $\mathrm{P}(E_3) = \frac{1}{4}$ we get

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

## 5.2 On the computational cost of the method

Let $c(n)$ denote the computational cost to compute the solution of a linear system with coefficient matrix either $L_{sym}^+$ or $Q_{sym}^-$. Standard iterative techniques allows to compute the smallest eigenvector of $L_{sym}^+$ or $Q_{sym}^-$ at a cost of $O(c(n))$ operations per step. We show that the use of Algorithms 2 and 3 allows to compute the eigenvectors of $L_{sym}^+ \# Q_{sym}^-$ with the same order of operations.

First of all it is important to realize that the matrix $H_s = V_s^T B V_s$, defined in line 5, can be defined iteratively and does not require any additional matrix multiplication [14]. Thus the cost of each iteration of Algorithm 3 dominated by lines 8 and 9, and requires $O(2c(n))$ ops. The algorithm converges exponentially, namely if $[a, b]$ is any interval containing the eigenvalues of $A^{-1}B$, then $\|\mathbf{x}_s - \mathbf{x}\| = O(\exp(-2s\sqrt[4]{a/b}))$, where $\mathbf{x} = (A^{-1}B)^{-1/2}\mathbf{y}$. See f.i. [14] for details. Thus $O(s_\varepsilon)$ iterations are enough to reach the prescribed tolerance $\varepsilon > 0$, where $s_\varepsilon = |\log \varepsilon / 2 \sqrt[4]{a/b}|$. However it is worth pointing out that in practice, at least for the matrices considered in this work, much less iterations than $O(s_\varepsilon)$ are enough. Therefore the proposed IPM technique allows to compute the smallest eigenvector of $L_{sym}^+ \# Q_{sym}^-$ at a cost of $O(c(n)) + O(2s_\varepsilon c(n))$ operations per step. This shows that the method is scalable. A final important remark concerns step 6. The matrix $H_s$ is tridiagonal of size $2s \times 2s$, thus the function $H_s^{-1/2}$ can be implemented directly using a method for dense matrices, without any notable change to the overall algorithm cost.

Next Figure 3 shows that, despite the computationally ugly definition of $L_{sym}^+ \# Q_{sym}^-$, we are able to compute its smallest eigenvector with a constant factor overcome, whereas the naive direct computation would be extremely prohibitive or unfeasible.

In Fig. 3 we show the median execution time for the computation of the smallest eigenvector of the signed ratio/normalized cut Laplacians , the balance ratio/normalized cut Laplacians and the geometric mean $L_{sym}^+ \# Q_{sym}^-$. We randomly generate graphs with a sparsity of 2.5% under the perfect stochastic case, *i.e.* $p_{in}^+ = p_{out}^- = 1$ and $p_{out}^+ = p_{in}^- = 0$, where the size of graphs goes from $10,000$ to $100,000$ in steps of $10,000$. For each setting we report the median execution time out of 10 runs. Experiments are performed using one thread.

For the computation of the smallest eigenvector of the signed ratio/normalized cut Laplacians and the balance ratio/normalized cut Laplacians we compute the Laplacian matrix (*i.e.* $L_{SR}$, $L_{SN}$, $L_{BR}$ and $L_{BN}$) and use the function `eigs` from Matlab. For the computation of the smallest eigenvector of the geometric mean we consider two approaches: one approach is based on the computation of the geometric mean $L_{sym}^+ \# Q_{sym}^-$ using the Matlab toolbox provided by [12] and then the use of the function `eigs` from Matlab (in Fig. 3 denoted as $L_{GM}(\text{eigs})$). The second approach is based on the Inverse Power Method of Algorithm 2 together with the extended Krylov method of Algorithm 3 (in Fig. 3 denoted as $L_{GM}(\text{ours})$).

We can see that the execution time of for signed Laplacians is rather similar. One can observe that the execution time for the geometric mean with Matlab's `eigs` is truncated for graphs that have more than 20,000 nodes. This happens as the computation of the geometric mean does not fit into memory. On the other side, the time execution for the geometric mean with the Inverse Power Method and extended Krylov methods (Algorithms 2 and 3) is comparable with the one of the signed Laplacians that use `eigs`. In particular it is noticeable that the time executions differs just by a constant factor.

# 6    Experiments