[Reviews · NeurIPS 2016]

Reviewer 1

Summary

This paper presents an argument for using the geometric mean of the normalized Laplacian matrix for each of the positive and negative normalized laplacian matrices associated with a signed graph. The authors present some evidence for this idea in terms of a stochastic block model. This is a small but nice observation. They show that the eigenvectors of simple block-models (e.g. matrices (pI + q*ones) and variations for the signed matrices). ) So the analysis is very simple for the highly idealized case. (There is no study of the empirical convergence to this distribution as found in Abbe et al.; https://arxiv.org/pdf/1405.3267.pdf) When this is used for spectral clustering, we need the eigenvectors of the geometric metean. They also show how the eigenvectors of the geometric mean might be computed more rapidly than forming the geometric mean and computing it's eigenvectors. This is based on a particular application of the extended Krylov subspace method that can rapidly compute various functions of matrices. This contribution is less clear (see below) due to some existing work on the area whose relationship the authors should clarify. The empirical results for real world datasets are fairly standard finding of "my new method is better". There are various parameters (e.g. # of classes) that could be varied (e.g. # of classes) and little sensitivity study, but this is not uncommon. The only troubling finding is on the Wikipedia signed network. The authors claim to have clustering structure based on a highly misleading plot, the dreaded "dot plot" which is arguably the weakest evidence possible. This section needs far more evidence to support its supposed conclusions. Questions for authors --------------------- 1. Could you clarify the relationship with other work on computing the geometric mean * a vector, e.g. * Krylov iterative methods for the geometric mean of two matrices times a vector Jacopo Castellini (too new to have been included as a ref.) * Computing the weighted geometric mean of two large-scale matrices and its inverse times a vector. http://eprints.ma.man.ac.uk/2474/01/covered/MIMS_ep2016_29.pdf (It seems this is related to [8], which you already cite.) In particular, what is the contribution of the Krylov algorithm in this paper vs. [8]. 2. Is there additional evidence for clustering in the wiki-sign network? Figs. 4(a) and 4(b) are not at all convincing. (Especially in light of multiple negative findings in [15] an and [4].) Suggestions for future work --------------------------- Given the results on the stochastic block model are largely expressed with the same subspaces (e.g. the eigenvectors are the same for the pos. and neg, it might be nice to study a more straightforward quantity such as A*B or B*A (or their sum) as a baseline measure to quantify the significance of your approach. This would also suggest a simple extension to your theorem about the blockmodel case for a broader class of matrix functions. Presentation remarks -------------------- * Harmless typos throughout that do not impede understanding * Theorems 1 and 2 are fairly simple and don't seem like they need to be highlighted. (And can probably be cited from existing work.)

Qualitative Assessment

This is a small but nice contribution. It seems like much of the theory could be generalized to a broader class of matrix functions (e.g. many things with the product of the Laplacians would seem to meet many of the criteria, so there is nothing special about the geometric mean). This could enable and allow better computational methods that are far more straightforward.

Confidence in this Review

3-Expert (read the paper in detail, know the area, quite certain of my opinion)


Reviewer 2

Summary

This paper aims to provide a notion of Laplacian on signed graphs that can be successfully used for clustering the nodes. This is done by considering the geometric mean of certain Laplacians defined on the positive and negative part of the network.

Qualitative Assessment

The paper is very well written and it is a pleasure to read. After a well written introduction the authors describe in section 2 the notion of geometric mean of two PSD matrices and list relevant properties of the construction. In section 4 the authors persuasively argue that standard definitions of Laplacian on signed graphs involving the average (arithmetic mean) of Laplacians on positive and negative parts of a graph are biased and in general do not lead to recovery of an underlying clustering structure. They do this by carefully analyzing a stochastic block model on signed graphs. In this same scenario thy show that they proposal, the LGM (geometric mean Laplacian), successfully captures the expected block structure under a certain regime. Section 4 contains the main theoretical contribution of this paper. In section 5 the authors point out that computing the geometric mean of two PSD matrices may in general be very expensive. Then they go on to propose a method for directly computing eigenvectors corresponding to small eigenvalues of the matrix geometric mean A#B of two given PSD matrices A and B. It is important that this is done without actually ever computing the metric A#B itself. Finally, section 6 of the paper contains some experiments. The experiment on the wikipedia dataset exemplifies that claim that the eigenvectors of LGM better capture clustering than those corresponding to LAM (arithmetic mean Laplacian on signed networks). Some notes/typos Line 101: Some condition on A is probably missing in the statement of the problem X^2 = A. Line 139: Define the function \lambda_i( ...) Line 324: less --> fewer Line 329 of Supp Mats: sensible -- > sensitive (?) Line 331 of Supp. Mats: overcome --> overhead (?)

Confidence in this Review

2-Confident (read it all; understood it all reasonably well)


Reviewer 3

Summary

In this paper, the authors propose a new approach for clustering signed networks. The usual approach is to apply spectral clustering methods to the sum of two signed laplacians (possibly normalized), corresponding to the negative ties and to the positive ties. The authors propose applying spectral clustering approaches to the geometric means of the two normalized Laplacians. The authors first justify the method by proving that under the stochastic block model, if the method is applied to the _expected_ laplacians, then the original clusters are recovered precisely for a larger range of parameters (in contrast with the usual approaches that have an inherent bias). The authors then propose a Krylov subspace + Conjugate Gradient based method for estimating the eigenvectors, and give experiments to indicate that it's comparable to the approaches for computing the eigenvectors of the usual signed laplacians. Finally, the authors apply this method to the Wikipedia adminship election dataset, and seem to recover a meaningful clustering where the previous approaches had failed. (The authors also report superior performance on several UCI datasets).

Qualitative Assessment

1. I quite liked the central idea of the paper, and demonstrating that for stochastic block models, the method recovers the block in the largest fraction of cases in comparison with the other methods. The case could be strengthened by proving that the method works for an interesting range of parameters for the stochastic block models with high probability, rather than just on the expected matrices. 2. The wikipedia result is certainly very interesting. It would be good to report the running times of the wikipedia experiments. 3. On UCI datasets, however, it seems that L+sym (rather than L_gm) seems to do best in all scenarios (from the figures in the supplementary material). Am I missing something here? 4. Since the matrices involved are laplacians, there are much better methods for solving systems in them rather than picking incomplete cholesky factorization -- search for algebraic mutligrid, or combinatorial multigrid. 5. Please provide additional specific justification for the claim that the reduction in the error for the EKSM is geometric (point to specific theorems in the reference) 6. I believe the signs in eq 3, def of the signed Laplacian are messed up. They should be D - W^+ + W^- 7. The text about the various conditions in table 1, and what they imply is confusing. It should be clarified if possible.

Confidence in this Review

2-Confident (read it all; understood it all reasonably well)


Reviewer 4

Summary

This paper proposed a geometric mean of two Laplacian matrices that are generated from a positive edge graph G+ and a negative edge graph G-, respectively. The authors further provide a fast algorithm to compute the geometric mean so that a spectral clustering can be applied on a signed network that consists of G+ and G-.

Qualitative Assessment

The idea of using a geometric mean of matrices, by my knowledge, is quit new. The proof of properties of matrix generated from a stochastic block model, defined in Line 190 from supplementary material is correct and the provided algorithm is reasonable. However, there are flaws that the authors could work on to improve their methods (Line numbers in the comment are based on the supplementary file since the summited paper is a shorter version with proof of theorem eliminated): 1. According to line 52 and Equation 3.7, 3.8 defined in reference [14], the author incorrectly implements the spectral clustering for L_{BN} and L_{SN}. From Eq(3) above line 88 and Eq(4) above line 92, the Laplacian must have non-positive L_{ij} except for the diagonal. However, Eq3.7 from reference [14] suggests that could have positive L_{ij} if node v_i and v_j has large negative edge weight. Just think of a simple case where nodes inside a cluster are fully connected with positive relation, but nodes in different clusters are fully connected with negative relation. W+ is an adjacency matrix with diagonal block of 1 and W- is an adjacency matrix with W-_{ij}=1 if v_i, v_j are in different clusters, L_{BR} _{ij}=-1 for i!=j. In this case, it is unsuitable to detect clusters with spectral clustering. This explains why methods with L_{BN} and L_{SN} have poor performance. The authors could correct their implementation for a fair comparison to their method. 2. In line 81, the authors mentioned that Q has zero eigenvalue and work if and only if the graph is bipartite. However, networks constructed from UCI dataset do not necessarily has bipartite components, which means Q^-_{sym} is not a reasonable baseline for comparison. The author could find dataset of bipartite graphs for evaluation. 3. There is issue of the proposed method according to the implementation in the experiment on UCI datasets. In an very simple case where each positive relation cancelled out the counter part negative relation between v_i and v_j, for all i and j, that is W+ = W-, a clustering algorithm should not be able detect any cluster. However, according to the geometric mean method, the method by the authors should incorrectly detect clusters defined by W+ or W-. For example, choose k+ = k- =N in the experiment on page 12. If the method detects well structured cluters with large noise of W-, then the result could be bias or overfitting. The author may think about the interpretation of the constructed Laplacian L_{GM} and give a reasonable explanation or solution to this issue. 4. Athougth W+ and W- defined in line 190 from SBM in Section 4 have the same eigenvector, they are special cases where the true clusters are already known. There is no guarantee that the sufficient and necessary condition that W+ *W- = W- *W+ hold on for the two matrix to have the same eigenvectors. It is unclear that how the author expand their discoveries to general cases to apply to spectral clustering.

Confidence in this Review

3-Expert (read the paper in detail, know the area, quite certain of my opinion)


Reviewer 5

Summary

The author want to cluster the node of a graph containing negative edges (which pull their endpoint in different cluster). They start by recalling various spectral formulations of the problem, showing that all combine the Laplacians of the positive and negative induced subgraphs. Then they claim combining those 2 Laplacians through geometric mean would better preserve the relative ordering of eigenvalues and thus provide more information in case of noise coming from one of the subgraph. They prove it is indeed the case in a simple Stochastic Block Model case (as confirmed by synthetic simulations) and provide a numerical scheme to efficiently compute the first eigenvector of the geometric mean A#B of two positive matrices A and B. Finally, they demonstrate their approach on real data, showing it outperforms other Laplacians methods.

Qualitative Assessment

Overall, the paper is well written and easy to follow. It presents an interesting idea which is well justified by intuition and rigorous linear algebra and provide interesting experimental results to a common problem of graph clustering (in the presence of negative information). Regarding the experiments, as opposed to the unsigned case where there exist graphs with known ground truth that are often used as standard benchmark, it is more difficult to find such graphs in the signed case. However, and despite its small size, it would be interesting to see the output of that method on the Gahuku-Gama Subtribes network [1] for instance. Anyway I think it's fair to use vectorial datasets and turn them into graphs yet it's not clear how is defined the clustering error that measures performance. If each class make one cluster, then why use k^+=10 in the iris case? One issue is that the proposed method doesn't show any improvement over the one using L^+_{sym} (though this might be due to the graph construction). This is also the case in the synthetic experiments presented in Figure 2, where it seems that either L^+_{sym} or Q^-_{sym} are better than the geometric means when there is positive or negative information respectively. The intuition about ordering of eigenvalues is interesting (although I think the fact that L^+ and Q^- have the same eigenvectors in the SBM case should be moved from the supplementary material to the main paper in the sake of clarity) but it should be commented how this extend to real graphs (where the spectrum of G^+ and G^- can be different). In section 2.2, I'm confused by equation (4): L_SR = L^+ + Q^-. If I understand correctly the definitions, L^+ = D^+ - W^+ and Q^- = D^- + W^- thus L^+ + Q^- = \bar{D} - W^+ + W^- ≠ \bar{D} - W^+ - W^- = L_SR Besides a new way of combining Laplacians and new structure exhibited in the Wikipedia dataset, it is not so common (although not novel) to see some guarantees when it comes to clustering, even if only for a regular SBM. Minor comments: l. 102: if that is possible, give a reference to say who introduce geometric mean of two matrices (I never came across that concept before) l. 152: “trough” l. 173: “the the” l. 187: in proof of corollary 1, it would be helpful to state the mapping between variables x_1 to x_4 and the various p^±_{out/in} l. 238: there seems to be enough space to at least give an indication about the overall asymptotic complexity of the algorithm l. 255: any reason to look for 30 clusters? If not, say so [1] Read, K. (1954). Cultures of the Central Highlands, New Guinea. Southwestern Journal of Anthropology, 10(1), 1-43. http://www.jstor.org/stable/3629074

Confidence in this Review

2-Confident (read it all; understood it all reasonably well)


Reviewer 6

Summary

This paper provides a new Laplacian matrix for signed network by taking the geometric mean of positive graph and negative graph. It proves that this Laplacian shares some common eigenvectors with the current Laplacian matrix under some constraints. It further discusses features of the eigenvectors under SBM model. It also gives a method for finding eigenvectors.

Qualitative Assessment

typo: equation (3), should be "+W^-“ The construction of the Laplacian is novel and interesting. The authors did a thorough study on this topic. I have 2 suggestions for the authors: 1. The motivation of this paper for using geometric mean is not very clear to me. It may be helpful if the authors can explain a little more about why geometric laplacian will outperform the arithmetic laplacian. 2. I am wondering how the N-cut of the geometric Laplacian will look like. It does not seem like geometric Laplacian will have a easy n-cut form to me. Since spectral clustering is closely related to minimizing n-cut, it would help if the authors could explain more.

Confidence in this Review

2-Confident (read it all; understood it all reasonably well)